# Sex Differences in Serious Adverse Events Reported Following Booster Doses of COVID-19 Vaccination in Thailand: A Countrywide Nested Unmatched Case-Control Study

**DOI:** 10.3390/vaccines11121772

**Published:** 2023-11-28

**Authors:** Chawisar Janekrongtham, Mariano Salazar, Pawinee Doung-ngern

**Affiliations:** 1Division of AIDS and STIs, Department of Disease Control, Ministry of Public Health, 88/21 Tiwanon Rd., Nonthaburi 11000, Thailand; 2Department of Global Public Health, Karolinska Institutet, Widerströmska huset Tomtebodavägen 18 A, Plan 3, 17165 Stockholm, Sweden; mariano.salazar@ki.se; 3Division of Epidemiology, Department of Disease Control, Ministry of Public Health, 88/21 Tiwanon Rd., Nonthaburi 11000, Thailand; pawind@gmail.com

**Keywords:** serious adverse events, COVID-19 vaccines, booster doses, sex difference, BNT162b2, mRNA-1273, ChAdOx1 nCoV-19, adverse events of special interest

## Abstract

A booster dose of a COVID-19 vaccine has been proven effective in restoring vaccine effectiveness and is currently recommended for use in some populations at risk of severe COVID-19 infection. Since sex differences in adverse events are significant in response to the vaccines, the safety of booster selection must be studied to avoid serious adverse events (SAE), such as life-threatening diseases. First, this study aimed to identify sex differences in SAE incidences using a prospective cohort design. Second, a nested unmatched case-control study was used to identify factors associated with reported SAE within 30 days after the booster shot. Multivariable logistic regression indicated the adjusted odds ratio by accounting for host and vaccine variables, thus, policy effects. The findings confirmed that SAE was rare and that age-sex-dominated disease classifications differed. Specific to SAE following the booster dose, we found that females aged 12–40 had a higher risk of being reported with SAE than males of the same age, while males over 50 had a higher risk than females. Other risk factors identified were the presence of metabolic syndrome and the use of certain vaccine brands. Mechanisms could be explained by individual host responses rather than the vaccines’ direct effect. Therefore, SAE could be preventable by age-sex-specific vaccine selection, post-vaccination precautions, and early symptom detection. Future vaccine development should aim to limit host-specific reactogenicity for safety concerns.

## 1. Introduction

Years after the coronavirus disease 2019 (COVID-19) was declared a global health threat in January 2020, the disease landscape has changed significantly after most of the world’s population has generated hybrid immunity through mass vaccination campaigns and/or infection-induced immunity. There is a general agreement that the primary series of COVID-19 vaccines is the most effective method for controlling the COVID-19 pandemic. Having been adopted in all countries worldwide, COVID-19 vaccines have saved millions of lives and reduced hospitalizations [1]. As of January 2023, the global vaccine coverage was 71%. The use of mRNA-based technologies, including BNT162b2 (Pfizer) and mRNA-1273 (Moderna), for the first time in vaccines, highlights the importance of a thorough study on its safety [2].

Sex differences in adverse events (AE) do exist [3,4,5,6,7]. Evidence suggests that females respond more effectively to COVID-19 vaccines and some viral vaccines, with a higher protection rate [3,4,5]. Females and young age are factors strongly associated with AE, but not specific to serious outcomes [6,7]. This finding applies to both primary series and booster doses of COVID-19 vaccination. Many underlying mechanisms, such as hormonal, physiological, and psychological, are believed to explain these different responses between sexes [4,8]. In regards to age, younger people are more likely to experience AE than older people because of their stronger immune response [9,10]. As a result, the risk–benefit of getting a boosting effect should be individually evaluated, especially considering their age and sex.

A booster dose for COVID-19 is defined as a dose administered to a vaccinated population that has completed two homologous or two heterologous primary vaccination series [11]. It is needed to restore vaccine effectiveness that diminishes with time [11]. The use of booster doses was more targeted towards high-priority groups, including older adults and those with underlying diseases, and was recommended to be given in 6–12 month intervals by the WHO [2] and the U.S. Centers for Disease Control and Prevention (CDC) [12]. The brand used for the booster dose can be similar to (homologous booster) or different from (heterologous booster) previous doses [13]. Both booster regimens are well-known and effective [14,15,16].

Unlike the second dose given several weeks after the first to ensure a stronger immune response during mass vaccinations, booster doses of the COVID-19 vaccine are given several months after an initial series of doses to maintain immunity in a more specific population [11]. Therefore, compared to the second dose of the primary series, booster doses have longer dose intervals and are given to hosts with lower immunity levels, which could lead to different host reactions. This highlights a need to conduct a study specifically on booster doses.

In Thailand, the national COVID-19 vaccination program has been operating since March 2021 [17]. The majority of initial vaccinations consisted of two doses of an inactivated vaccine, CoronaVac (Sinovac), for those aged 18–59 years or two doses of viral vector-based vaccine, ChAdOx1 nCoV-19 (AstraZeneca), for those aged above 60. Under this national scheme, residents in Thailand were able to be vaccinated free of charge [14,18]. 

In response to vaccine shortages, Thai authorities recommended heterologous primary series vaccination with Sinovac and AstraZeneca as evidence showed high immunogenicity and safety based on real-world data in the Thai population [19]. The heterologous booster doses policy in Thailand started on 7 July 2021 with AstraZeneca being the first booster vaccine adopted. Pfizer and Moderna were available as booster choices later in Thailand, from 9 August 2021 and 1 November 2021, respectively. The timeline for the vaccination rollout can be seen in Figure 1. 

Since some of these vaccines were made with new technologies, were globally distributed, and exhibit sex-specific responses, safety brand selection must be studied to avoid serious adverse events (SAE), including those who died, had life-threatening diseases, such as myocarditis and myocardial infarction, or were permanently handicapped (Appendix A). No studies to date have investigated sex differences, specifically in SAE following booster doses. Additionally, the situation in Thailand is unique as multiple heterologous primary series and booster regimens have been used to tackle vaccine shortages. This presents an opportunity to investigate the SAE. following each heterologous booster regimen. Information like this may offer flexibility regarding vaccine acceptance, supply, or availability in Low- or Middle-Income Countries (LMICs) with the possibility of minimizing AE. This study aims to examine sex differences in SAE incidences and factors associated with SAE reported after a first booster dose of COVID-19 vaccination in Thailand.

## 2. Materials and Methods

### 2.1. Study Designs and Study Populations

Two study designs were used to answer the research questions. First, a large prospective observational cohort study was conducted to describe the sex differences in incidences of SAE reports in Thailand after receiving any booster dose of COVID-19 vaccination through the national Adverse Event Following Immunization surveillance system under the Department of Disease Control (AEFI-DDC). This study included reports of those aged ≥12 who received a booster vaccine from one of the three most used vaccine brands, Pfizer, AstraZeneca, and Moderna, since the beginning of the booster policy on 7 July 2021 until 31 December 2022. Exclusion criteria were people with a symptom onset of SAE for more than 30 days. 

Secondly, an unmatched nested case-control study was performed among a cohort of individuals who received a first booster dose to evaluate factors associated with having SAE. We selected only SAE reporting after the first booster dose because differences in total doses could affect the analysis and interpretation. They have been given to different populations. Cases were defined as people with a first booster dose who presented with any SAE in the AEFI-DDC database from 7 July 2021 to 31 December 2022. A control was defined as people with a first booster dose who did not report SAE but got vaccinated during the same period from the Ministry of Public Health Immunization Center Program (MOPH-IC) database.

Four times as many controls were randomly selected from the same source population compared to the number of SAE cases. Before sampling, hashed card identification was used to identify the cases within the source population and they were excluded from the sampling frame. The sampling method was stratified proportionately to the brand of vaccine distribution in the source population.

### 2.2. Data Extraction and Management

Two national databases, the MOPH-IC and AEFI-DDC [21], were used for data extraction. The MOPH-IC database collects vaccine administration information from the beginning of the COVID-19 vaccination campaign. It collects data from public and private providers regarding national identification numbers, risk categories, including major underlying conditions (such as obesity, chronic lung diseases, diabetes mellitus, and cancer), occupations at risk of COVID-19 exposure, vaccination dates, and vaccine brands.

The AEFI-DDC database monitors SAE following immunization using data from public and private hospitals in Thailand. It is a passive surveillance report system set up over 20 years ago and has been recently updated from a paper fax system to an online one during COVID-19. Compared to private hospitals, the program is more established in public hospitals, which account for 75% of total hospitals and 79% of total hospital beds in Thailand [22]. 

SAE was determined based on the doctor’s diagnosis at the hospital. Hospital staff investigated all events reported and defined them as SAE cases if they met eligibility criteria (Appendix A). All events were filed through an online form and reported to the database. The information collected for SAE cases included presenting symptoms, date of onset, date of admission, treatment status, related laboratory results, underlying diseases, provisional diagnosis, and conclusions made by physicians in charge at the hospitals. All people who died before arriving at hospitals with unknown causes or unattended deaths were also investigated. Here, we used the doctor’s diagnosis as the final diagnosis regardless of the causality assessment. If an AEFI report had more than one booster vaccination entered and can indicate the difference in the order of doses, then all records were kept in the analysis. 

### 2.3. Study Definition

Each AEFI report is defined as a person who had an onset of SAE within 30 days after the vaccination according to the program criteria (Appendix A). SAE was classified into 5 most common categories using the Pharmacovigilance Systems in the European Union and the United States as follows [23]: (1) cardiovascular events, such as acute myocardial infarction, heart failure, myocarditis, and cardiac arrhythmia; (2) thrombotic events, such as ischemic stroke, pulmonary embolism, and venous thrombosis; (3) allergic events, such as anaphylaxis and Steven–Johnson syndrome; (4) neurological events, such as neuropathy, neuritis, and unspecified stroke; (5) hemorrhagic events, such as hemorrhagic stroke, brain hemorrhage, and gastrointestinal bleeding. Any events beyond these five categories were classified as “others”. Deaths were defined as individuals who reported fatal outcome after treatment. Deaths with unknown causes were placed in the “others” group.

The main exposure of interest was sex, classified as male or female, based on a biological distinction from national data registration. Potential confounders that could be associated with reported SAE were explained by three categories (Figure 2): (1) host variables, including age and comorbidities; (2) vaccine variables, including brands of a booster dose, regimens of a booster dose (Appendix A), and booster dose interval; and (3) variables that affect reporting include occupation and effect of policies.

### 2.4. Data Analysis

A complete case analysis was used to handle missing data on exposure and outcome of interests, while other variables were kept with presenting the percentages of missing values. All statistical analyses were performed using STATA^®^ version 17. Descriptive statistics were used to describe characteristics, including frequency, proportion, and median with interquartile range (IQR). Between-group comparisons were carried out using Pearson’s Chi-square test or the Wilcoxon rank-sum test. Moreover, each potential confounder was first examined to see if it was related to the exposure and outcome before being used to build adjusted models. A *p*-value of less than 0.05 was considered a statistically significant difference. If variables indicated differences, subgroup comparisons were further conducted. 

Odds ratios and 95% confidence intervals (95% CI) were calculated by logistic regression. The multivariable analysis started with a full model with all variables based on the theoretical framework that entered the model simultaneously. Then, a purposeful selection algorithm was applied to select variables. Therefore, covariates were removed from the model if they had a *p*-value of more than 0.1 in the univariate model and were not a confounder. 

Multicollinearity was evaluated. Interaction terms were created and evaluated to determine whether variables had an interaction. Then, stratum-specific association between interacting variables was interpreted. Additionally, a quadratic prediction model of reported SAE using age (x) and quadratic of age (x^2^) was conducted by visually inspecting the model’s predictions on the interactions for the best data fitting.

## 3. Results

During the study period, 33,394,922 (99.2% of the target population) booster doses met the study’s eligibility criteria (Appendix A) and were defined as the source population (Figure 3). The most used brand was Pfizer at 65.5%, followed by AstraZeneca at 17.5%, and Moderna at 17.0%., Vaccine distribution brands differed by age group (Appendix A). The male-to-female ratio was 1:1.3, and the age group distribution in the population was ages 12–17 (3.0%), ages 18–20 (2.6%), ages 21–40 (37.6%), ages 41–60 (37.4%), ages 61–80 (17.6%), and ages above 80 (1.8%).

### 3.1. Sex Differences in Incidences of SAE Reported Following Immunization

Of the 2600 events reported following booster doses through the AEFI-DDC system between 7 July 2021 and 31 December 2022, 590 (22.7%) were eligible for SAE definition. Then 552 (93.6%) events were included because they had onset within 30 days following any booster doses (Figure 3). Two of the 552 cases had COVID-19 infection during the same reported period. Of these cases, 490 (88.8%) events were reported after the first COVID-19 booster dose, 60 (10.9%) after the second, and 2 (0.4%) after the third. 

Table 1 presents characteristics of the source population who presented with SAE stratified by sex. There were more females than males. Conversely, deaths were higher in males than in females. The median age was 53 (Q1, Q3: 39, 68), and males were statistically older than females. There was no difference between males and females reporting underlying diseases, the brand, and the regimen of booster dose used. More cases presented with systemic reactions than local reactions. Females presented with a greater proportion of reactogenicity in both systemic and local presenting symptoms than males with statistical significance. The overall median onset of SAE was two (Q1, Q3 = 0, 8) days after vaccination, whereas males presented with longer periods than females.

There was a difference in SAE classification between females and males: Males mainly presented with cardiovascular events (n = 101, 38.5%) followed by thrombotic events (n = 70, 26.7%), while females mainly presented with thrombotic events (n = 73, 25.2%) followed by allergic events (n = 57, 19.7%) and cardiovascular events (n = 55, 19.0%). Cardiovascular and allergic events were statistically significant differences across the sexes. Males aged 61–80 presented with SAE more than females at the same ages; conversely, females presented with SAE more than males aged 21–40, with statistical support.

For incidences, males reported 17.9 events per million doses, and females reported 15.5 events per million doses administered (Table 2). Incidences were insignificant differences by sex; they are described in Table 2. Most reporting followed the first booster vaccination (18.8 events per million doses administered). The fatality rate was 4.6 per million doses provided with males dominated.

AstraZeneca had contributed to maximum incidence across most age groups in both sexes except one above 80 years old (Table 3). The older the age, the higher incidences of SAE were noted in the same direction (Table 3). However, the youngest males aged 12–17 and males aged 61–80 reported SAE following Pfizer showed the opposite trend, but no statistical difference compared to females of the same age. On the contrary, the oldest female aged above 80 had remarkably higher incidences of SAE following AstraZeneca and Moderna (Table 3).

### 3.2. Sex Difference in Factors Associated with Reported SAE Following the First Booster Dose

There was recruitment of 490 cases and 1960 controls for the analysis as the study population (Figure 3). The proportions of sexes, age groups, and booster vaccination brands were equivalent between the source and control populations. Appendix A indicates that the distribution of sexes among cases and controls was similar. However, the median age of cases (55 years) was 11 years older than that of controls (44 years). Cases reported more frequently receiving AstraZeneca, having metabolic syndrome, working as frontline workers, receiving heterogeneous booster dose, reporting within three months after booster dose policy initiation, and having a dosing interval of fewer than three months with a significant difference (Appendix A). 

Even though being female had a non-significant association with reported SAE in univariable analysis, the adjusted model had a significant protective effect (Adjusted OR = 0.79, 95% CI = 0.58, 0.98) (Table 4). Older ages above 50 (vs. aged 31–40) and ones with metabolic syndromes (vs. no reported) were at risk of reporting SAE. Furthermore, an interaction between age and sex was observed; hence, age stratification by sex was used to explain this association. Quadratic line graphs in Figure 4 predicted that females aged less than 40 (12–40) had a probability of reporting SAE higher than males at the same ages. In contrast, males aged more than 50 years had a probability of reporting SAE higher than females at the same ages with statistical significance (turning point 45 years old). Therefore, the probability of reported SAE after booster doses was the highest among older males (above 45), followed by older females (above 45), younger females (12–45), and younger males (12–45). 

Considering the disease classifications, we found that the two most common disease classifications of males similar between older and younger groups include cardiovascular events (older 37.8%, younger 39.0%), followed by thrombotic events (older 30.3%, younger 27.0%). In contrast, the third most common was reported differently in the older and younger groups, hemorrhagic events (6.5%), and neurological events (15.3%), respectively. For females, the most common classifications were thrombotic, cardiovascular, and allergic events in both the older and younger groups, however, with a different ordering. The maximum proportion reported among younger females was allergic events (27.7%), while among older females, it was thrombotic events (32.6%).

Brands of COVID-19 vaccine significantly associated with the reported SAE AstraZeneca (vs. Moderna) (aOR = 5.97, 95%CI = 2.52, 14.13), receiving Pfizer (vs. Moderna) (aOR = 3.16, 95%CI = 1.76, 5.69). The subgroup analysis of sex in the interaction model indicated a notable difference in the risk of SAE between AstraZeneca and mRNA-based vaccines in younger females than in other groups (Appendix A). Moreover, the heterogeneity of booster regimens had no association with the outcomes. The dosing interval of more than three months also has no association in the multivariable model. 

The adjusted Odds ratios of reporting SAE were highest during the first three months compared to nine months after the booster dosage policy was implemented (aOR = 7.65, 95% CI = 3.05, 19.19), and after the period following booster vaccine introduction (aOR = 3.75, 95% CI = 1.38, 10.14). Additionally, working as frontline workers (aOR = 2.31, 95%CI = 1.54, 3.46) had a positive association. Additionally, working as frontline workers (aOR = 2.31, 95%CI = 1.54, 3.46) had a positive association.

## 4. Discussion

Incidences of SAE reported 30 days following a booster dose of a COVID-19 vaccination were rare among the population in Thailand, where the majority had been vaccinated with Pfizer vaccine, followed by AstraZeneca and Moderna. Sex differences in response to the booster doses of a COVID-19 vaccination and age-sex-dominated disease classifications differed. Specific to SAE following the booster dose, we found that risks are generally higher in males, but not always. Other risk factors were the presence of metabolic syndromes and the use of certain vaccine brands. The study concluded that the age groups at high risk of reported SAE after receiving a booster dose of a COVID-19 vaccination were older males, older females, and younger females. 

Similar trends were published regarding incidences of adverse events by the number of doses. Many publications supported that AE caused by booster doses were less severe and had lower SAE incidence [7] compared to the second dose of the primary series [24,25]. The incidences did not rise by increasing the number of booster doses [26,27]. Moreover, sex differences across disease classification were similar to previous studies, including allergic events dominating in females [28] and cardiovascular events dominating in males [29].

This study also found that age modified the effect of sex. Females aged 12–40 had a higher risk of being reported SAE than males of the same age, while males over 50 had a higher risk than females. The risk in older people increased rapidly, followed by increasing age. Regardless of serious outcomes, most studies reported the opposite trend: females have a higher risk than males, and younger adults have a higher risk than older adults [3,4,5,6,7,8]. For SAE after the primary doses, some conditions dominated in younger individuals, such as myopericarditis [30,31,32] and anaphylaxis [4,28], which was similarly found in this study after booster doses. This study further evaluated the risk of all SAE, in which half of the cases were cardiovascular events (other than myopericarditis) and thrombotic events. Since these diseases are commonly found in the older age group due to aging processes or their underlying conditions, it could be a reason why this study found the risk dominated in older people. Moreover, the older age group had more systemic side effects following vaccination than the younger group, which could be a supporting mechanism precipitating underlying diseases [33]. Therefore, the evaluation of overall SAE could be capturing immune-mediated reactions [34] after booster dose immunization. Nevertheless, without a causality assessment, the result was possibly an overreporting of SAE among the elderly.

Higher cardiovascular (CVD) risks [35] and faster degenerative changes [36] are the possible pathophysiological mechanisms that explain why older males were more likely than older females to be triggered by reactogenicity after vaccination, resulting in SAE. The reasons for the higher CVD risks among males include having more risky behaviors, such as smoking and drinking alcohol [35], being unlikely to go for a regular check-up, being late presentation to the hospitals [37], and being less likely to engage in protective health practices [38]. These supports the finding in our study that the effect of aging was multiplied by increasing age, and those with any metabolic syndrome were combined at risk of the SAE reported. Compared to older males, older females presented with dominantly thrombotic events, followed by cardiovascular events, which could be attributed to sex hormones [39,40].

A stronger immune response and female sex hormones could explain why younger females had a higher risk of being reported to have SAE after a booster dose compared to younger males [3,4,5,6,7]. The overproduction of inflammatory cytokines in females after the vaccination [41] is correlated to the finding that females have higher reactogenicity and dominantly allergic events. At the same time, thrombosis in multiple organs is attributed to female sex hormones [39,40]. However, the events among young females could be misclassified, and the risk could be overestimated. For instance, immunization stress-related responses (ISRRs), commonly seen in stroke-mimic symptoms in young females [42], could be overdiagnosed, and the vaccine campaign primarily faced this difficulty in vaccine acceptance [43]. 

On the other hand, the risk of SAE following a booster dose in the younger males could be underestimated due to rare younger males receiving a booster dose. Compared to a countrywide study that used the same databases of primary doses, there was a larger study population of young males that made it possible to capture 137 cases of myocarditis and pericarditis and report the highest risk in young males [30]. The symptoms of myopericarditis in the young were mild, and they recovered without permanent heart damage [30,31,32]. Therefore, the staff reporting AEFI-DDC surveillance system that was designed to capture SAE could have missed these cases. However, this study found a notable incidence of SAE among males aged 12–17 who received Pfizer as a booster. Hence, these specific groups should not be ignored.

SAE could be preventable by limiting the consequences of an excessive immune response stage after immunization, especially for the high-risk groups mentioned above. Approximately two-thirds of vaccinated individuals with reported SAE presented with systemic symptoms. Avoiding all activities that may increase inflammatory cytokines, such as vigorous exercise, alcohol consumption, cigarette smoking, and hot baths for approximately one week after vaccination [44], should be recommended. For example, Singapore’s policy advises adolescents and young adults to avoid exercise or strenuous physical activity for one week after vaccination [45]. In addition, vaccination should be delayed after getting COVID-19 infected for three months [46] to prevent SAE [47,48]. Therefore, individuals with high leftover antibodies (higher immunogenicity), either from infection or vaccine, should avoid getting the vaccination.

AstraZeneca contributed to a higher risk of reported SAE following a booster dose compared to mRNA-based vaccines while also being the only brand that contributed to a higher incidence among females aged 21–40 compared to males of the same age in this study. A higher incidence of SAE following AstraZeneca was correlated with more systemic side effects, such as fever and headache compared to Pfizer [49]. As a result, these symptoms could exacerbate underlying diseases, illustrating the importance of communicating age-sex-specific warning symptoms. Healthcare workers must observe symptoms after vaccinations and take prompt actions to relieve symptoms, preventing them from worsening. Some recent publications also reported the highest incidence of SAE in AstraZeneca [26], risk of cardiac-related deaths [50], and risk of severe thrombotic events [51,52]. As a result, in 2023, AstraZeneca is no longer used in the United Kingdom [53]. For safe vaccine selection, high-risk groups, for example young females should be informed about the possible adverse events and how to take care of themselves after vaccine administration. 

The heterologous booster regimen was determined to be safe from SAE by some publications but can increase AE risks compared to the homologous booster regimen [16,54]. Therefore, it should be recommended to support a supply chain interruption for its effectiveness [14,15,16] and safety [55]. According to previous observational studies [56,57], to provide robust immunogenicity for the boosting effect of the vaccine and to minimize AE [32], intervals between booster vaccinations should be extended.

Thailand’s vaccine policies and female-dominated frontline workers were confounded, affecting the results of this study. People were more likely to report SAE when the government announced the new vaccine policies; however, as they gained a higher level of trust in vaccine safety over time due to the availability of safety data, the policies ’effects faded over time. By adjusting them, the study could be more precise by limiting some reporting biases.

The findings are subject to at least five limitations. First, there are possible underreported cases for two reasons. Some people seek medical care in private hospitals, which have limited reporting in the surveillance system of AEFI-DDC. Moreover, pandemic fatigue could result in underreporting among the second and third booster doses. Second, underreported underlying comorbidities could happen because the MOPH-IC program is not designed to record all underlying diseases or multiple risks. Nonetheless, the associations were robust to change to exclude frontline workers from the study population. Third, because of the limitations of the databases, it was not possible to link the history of COVID-19 infection in the study population, given that recent infection could be associated with SAE and should be adjusted. Fourth, sparse data bias could happen in the >81 years age group due to a small sample. This could result in opposite trends of the incidence of SAE across sexes (Table 3) compared to the 61–80 age group, but no statistically significant sex differences occurred in the >81 years age group (Table 1). Lastly, causality could not be assumed due to an inability to confirm diagnoses. It is important to note that these reports should not be interpreted as causally related to the vaccines. Despite these limitations, the findings are internally valid because systemic errors are addressed through the study’s design, analysis, and interpretation. It could be valuable information for settings with similar vaccine regimens and policies. Before countries adopt new booster dose policies for specific populations, research on its incremental benefit should be conducted. 

## 5. Conclusions

SAE after booster doses were rare in Thailand. The findings first provided a comprehensive picture of the sex-modifying risk of reported SAE following a booster dose across ages. Sexes responded differently to the vaccines. Mechanisms could be explained by individual host response rather than vaccines’ direct effect, and as a result, SAE could be preventable. Therefore, pre-vaccination age and sex considerations and post-vaccination precautions should be made to limit serious consequences. More research on alternative booster choices, specific adverse events among those with metabolic syndromes, and countries’ cost-effectiveness analysis should be carried out.

## Figures and Tables

**Figure 1 vaccines-11-01772-f001:**
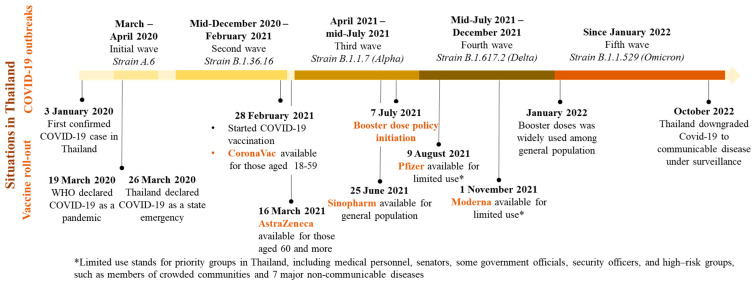
COVID-19 situation and vaccination rollout timeline in Thailand [17,20].

**Figure 2 vaccines-11-01772-f002:**
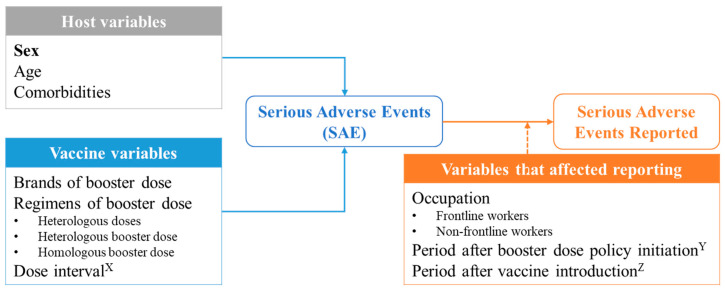
Conceptual framework of explanatory variables of reported SAE following a first booster dose of COVID-19 vaccination in Thailand. ^X^Dose interval: Time between the date of booster vaccination and the date of second dose vaccination in days. ^Y^Period after booster dose policy initiation: Time between the booster vaccination date and the policy announcement date in days. ^Z^Period after vaccine introduction: Time between the booster vaccination date and the date of each vaccine availability in days.

**Figure 3 vaccines-11-01772-f003:**
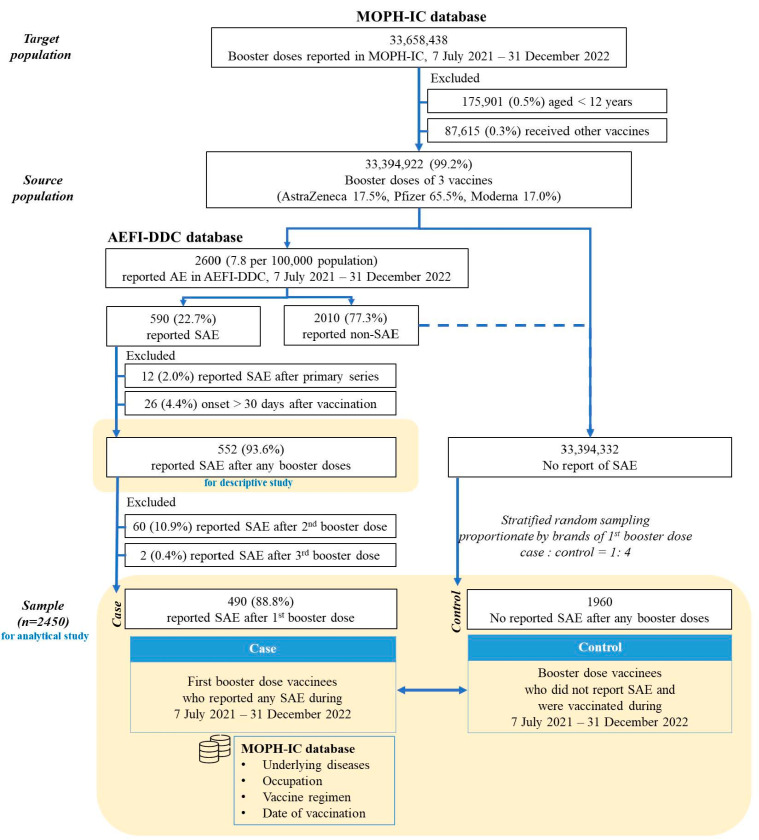
Study population.

**Figure 4 vaccines-11-01772-f004:**
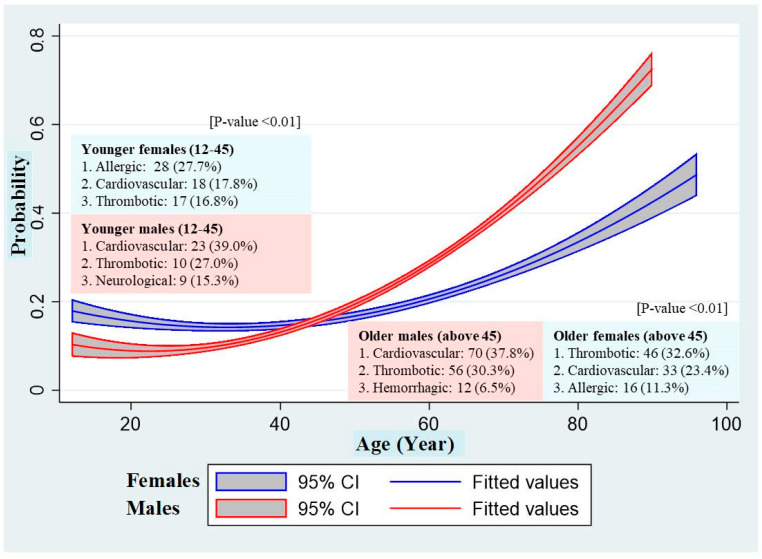
Probability of reported SAE after a booster dose of COVID-19 vaccination from the adjusted model with interactions across age stratified by sex and the most common SAE classification in each subgroup (n = 2450).

**Table 1 vaccines-11-01772-t001:** Characteristics of reported serious adverse events stratified by sex (n = 552).

Variables	Categories	Total (n = 552)	Male (n = 262, 47.5%)	Female (n = 290, 52.5%)	*p*-Value
Death outcomes	Yes	152 (27.5%)	98 (37.4%)	54 (18.6%)	<0.01
	No	400 (72.5%)	164 (62.6%)	236 (81.4%)	-
Age, median year (IQR)	Cont.	53 (39, 68)	59 (45,70)	48 (35, 63)	<0.01
Age group (years)	12–20	18 (3.3%)	8 (3.1%)	10 (3.4%)	0.78
	21–40	130 (23.7%)	42 (16.0%)	88 (30.3%)	<0.01
	41–60	202 (36.9%)	90 (34.4%)	112 (38.6%)	0.27
	61–80	174 (31.8%)	113 (43.1%)	61 (21.0%)	<0.01
	81+	24 (4.4%)	8 (3.1%)	16 (5.5%)	0.15
	missing	4 (0.0%)	1 (0.4%)	3 (1.0%)	-
Underlying diseases	Metabolic diseases	62 (11.2%)	34 (13.0%)	28 (9.6%)	0.40
	Others	15 (2.7%)	8 (3.1%)	7 (2.4%)	-
	No reported	475 (86.1%)	220 (84.0%)	255 (87.9%)	-
Number of booster doses	1	490 (88.8%)	245 (93.5%)	245 (84.5%)	<0.01
	2	60 (10.9%)	16 (6.1%)	44 (15.2%)	<0.01
	3	2 (0.4%)	1 (0.4%)	1 (0.3%)	0.94
Brands of booster dose	AstraZeneca	143 (25.9%)	66 (25.2%)	77 (26.6%)	0.72
	Moderna	55 (10.0%)	28 (10.7%)	27 (9.3%)	0.59
	Pfizer	354 (64.1%)	168 (64.1%)	186 (64.1%)	1.00
Regimens of booster dose	Heterologous doses	57 (10.3%)	29 (11.1%)	28 (9.7%)	0.39
	Heterologous booster dose	347 (62.9%)	160 (61.1%)	187 (64.5%)	-
	Homologous booster dose	148 (26.8%)	73 (27.9%)	75 (25.9%)	-
Presenting symptoms ** one person can have both*	Systemic reactions *	350 (63.4%)	150 (57.3%)	200 (69.0%)	<0.01
	Local reactions * missing	43 (8.0%) 191 (34.6%)	9 (3.4%) 110 (42.0%)	34 (11.7%) 81 (27.9%)	<0.01 -
Onset duration, median day (IQR)	Cont.	2 (0, 8)	4 (1, 11)	1 (0, 6)	<0.01
SAE classification	Cardiovascular	156 (28.3%)	101 (38.5%)	55 (19.0%)	<0.01
	Thrombotic	143 (25.9%)	70 (26.7%)	73 (25.2%)	0.68
	Allergy	67 (12.1%)	10 (3.8%)	57 (19.7%)	<0.01
	Neurological	43 (7.8%)	19 (7.3%)	24 (8.3%)	0.65
	Hemorrhage	31 (5.6%)	16 (6.1%)	15 (5.2%)	0.63
	Others/unclassified	112 (20.3%)	46 (17.6%)	66 (22.8%)	0.13

Data are n (%), unless otherwise indicated; Test for *p*-values: Pearson’s chi-squared test for categorical variables, Wilcoxon rank-sum test for continuous variables. * one person can have both.

**Table 2 vaccines-11-01772-t002:** Incidences of serious adverse events reported after booster doses by number of booster doses, death, and common SAE classification stratified by sex (n = 552).

Variables	Total		Male (N = 14,714,963)		Female (N = 18,731,694)		*p*-Value
	Events	Events per Million Doses	Events	Events per Million Doses	Events	Events per Million Doses	
All (N = 33,394,922)	552	16.5	262	17.9	290	15.5	0.87
**Death**	152	4.6	98	6.7	54	2.9	0.61
Number of booster doses							
- 1 booster (N = 26,052,993)	490	18.8	245	21.2	245	16.9	0.80
- 2 boosters (N = 6,345,264)	60	9.5	16	6.0	44	12.0	0.81
- 3 boosters (N = 996,665)	2	2.0	1	2.5	1	1.7	0.98
SAE classification							
- Cardiovascular events	156	4.7	101	6.9	55	2.9	0.56
- Thrombotic events	143	4.3	70	4.8	73	3.9	0.90
- Allergic events	67	2.0	10	0.7	57	3.0	0.64
- Neurological events	43	1.3	19	1.3	24	1.3	1.00
- Hemorrhagic events	31	0.9	16	1.1	15	0.8	0.93

**Table 3 vaccines-11-01772-t003:** Incidence (per million doses administered) of serious adverse events reported after booster doses by the vaccine brands and age groups stratified by sex (n = 552, N = 33,394,922).

Age	AstraZeneca (N = 5,834,495)			Moderna (N = 5,692,599)			Pfizer (N = 21,867,828)			All Vaccines (N = 33,394,922)		
	Male	Female	*p*-Value	Male	Female	*p*-Value	Male	Female	*p*-Value	Male	Female	*p*-Value
12–17 y (N = 1,006,391)	NA	NA	NA	0.0 (0/7403)	0.0 (0/8127)	1.00	14.7 (6/408,290)	8.6 (5/581,371)	0.93	14.4 (6/416,359)	8.5 (5/590,032)	0.93
18–20 y (N = 871,913)	18.6 (1/53,753)	17.8 (1/56,218)	1.00	0.0 (0/48,882)	0.0 (0/69,556)	1.00	3.7 (1/268,885)	10.7 (4/374,619)	0.92	5.4 (2/371,520)	10.0 (5/500,393)	0.94
21–40 y (N = 12,580,785)	10.7 (12/1,121,215)	24.6 (30/1,220,237)	0.80	5.4 (6/1,115,231)	7.6 (12/1,588,293)	0.95	7.2 (24/3,321,928)	10.9 (46/4,213,881)	0.87	7.6 (42/5,558,374)	12.5 (88/7,022,411)	0.79
41–60 y (N = 12,503,482)	21.2 (25/1,178,235)	21.4 (31/1,447,884)	1.00	13.7 (12/877,183)	5.9 (7/1,180,272)	0.86	15.6 (53/3,397,340)	16.7 (74/4,422,568)	0.97	16.5 (90/5,452,758)	15.9 (112/7,050,724)	0.98
61–80 y (N = 5,892,310)	83.9 (26/309,978)	31.1 (12/385,390)	0.77	28.5 (9/315,731)	9.6 (4/418492)	0.85	38.8 (78/2,007,848)	18.3 (45/2,007,848)	0.68	42.9 (113/2,633,557)	18.7 (61/3,258,753)	0.59
>80 y (N = 540,041)	37.8 (1/26,485)	88.5 (3/33,900)	0.94	39.9 (1/25,081)	104.3 (4/38,348)	0.93	33.5 (6/179,094)	38.0 (9/237,133)	0.98	28.3 (8/230,660)	51.7 (16/309,381)	0.90
All ages	24.5 (66/2,690,332)	24.5 (77/3,144,163)	1.00	11.7 (28/2,389,511)	8.2 (27/3,303,088)	0.89	17.4 (168/9,583,385)	15.1 (186/12,284,443)	0.89	17.9 (262/14,663,228)	15.5 (290/18,731,694)	0.87

Color code- white: <1.0, yellow: <10.0, pale orange: <20.0, dark orange: <35.0, pale red: <80.0, red: ≥80.0 per million doses administered missing age = 4 observations. NA-AstraZeneca was not recommended in Thailand for use in people aged 12–17; however, some unintentional misuse occurred including 666 doses in males and 534 doses in females.

**Table 4 vaccines-11-01772-t004:** Univariable and multivariable logistic regression analysis to estimate the odds of reported serious adverse events following booster dose of COVID-19 vaccination (n = 2450).

Variables	Crude Odds Ratio	[95% Conf	Interval]	*p*-Value	Adjusted Odds Ratio *	[95% Conf	Interval]	*p*-Value
Sex				0.06				
Females	0.83	0.68	1.01	0.06	0.79	0.63	0.98	0.03
Males	reference	-	-	-	reference	-	-	
Age group				<0.01				
12–20 years	1.00	0.56	1.80	1.00	1.41	0.72	2.76	0.31
21–30 years	1.13	0.75	1.69	0.57	1.13	0.72	1.76	0.60
31–40 years	reference	-	-	-	reference	-	-	-
41–50 years	1.40	0.96	2.04	0.08	1.17	0.77	1.78	0.45
51–60 years	2.26	1.56	3.28	<0.01	2.27	1.50	3.41	<0.01
>60 years	3.47	2.48	4.86	<0.01	5.00	3.45	7.27	<0.01
Brands of booster dose				<0.01				
AstraZeneca	3.27	2.24	4.78	<0.01	5.97	2.52	14.13	<0.01
Moderna	reference	-	-	-	reference	-	-	-
Pfizer	1.97	1.39	2.78	<0.01	3.16	1.76	5.69	<0.01
Underlying diseases				0.07				
Metabolic syndromes	1.46	1.07	2	0.02	2.07	1.46	2.94	<0.01
Other diseases	1.08	0.58	2.01	0.80	1.29	0.67	2.49	0.44
Unreported	reference	-	-	-	reference	-	-	-
Occupation				<0.01				
Frontline workers	3.61	2.67	4.89	<0.01	2.31	1.54	3.46	<0.01
Non–frontline workers	reference	-	-	-	reference	-	-	-
Regimens of booster dose				0.07				
Heterologous doses	0.98	0.72	1.35	0.91	1.04	0.66	1.66	0.85
Heterologous booster dose	1.25	0.95	1.65	0.11	1.04	0.68	1.61	0.85
Homologous booster dose	reference	-	-	-	reference	-	-	-
Dose interval				<0.01				
<3 months	reference	-	-	-	reference	-	-	-
3–6 months	0.49	0.37	0.64	<0.01	1.05	0.72	1.53	0.79
>6 months	0.54	0.37	0.77	<0.01	1.67	0.97	2.85	0.06
Period after booster dose policy initiation				<0.01				
<3 months	14.16	8.43	23.79	<0.01	7.65	3.05	19.19	<0.01
3–6 months	1.62	1.15	2.27	<0.01	1.48	0.84	2.62	0.17
6–9 months	1.09	0.81	1.48	0.57	0.97	0.62	1.52	0.89
>9 months	reference	-	-	-	reference	-	-	-
Period after vaccine introduction				0.26				
<3 months	1.24	0.85	1.81	0.26	3.75	1.38	10.14	0.01
3–6 months	1.20	0.91	1.57	0.20	1.78	0.95	3.31	0.07
6–9 months	0.97	0.72	1.31	0.86	1.17	0.72	1.90	0.53
>9 months	reference	-	-	-	reference	-	-	-

*p* < 0.05: statistically significance; * Adjusted for age, vaccine brand, underlying diseases, occupation, vaccine regimen, dose interval, period after booster dose policy initiation, and period after vaccine introduction.

## Data Availability

The data presented in this study are available on request from the corresponding author.

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
