# Peer review of "Sex Differences in Serious Adverse Events Reported Following Booster Doses of COVID-19 Vaccination in Thailand: A Countrywide Nested Unmatched Case-Control Study"

_vaccines, 2023, doi:10.3390/vaccines11121772_

Round 1

Reviewer 1 Report

Comments and Suggestions for Authors

In this manuscript, the authors employed a two-prong study design approach to investigate gender differences in serious adverse effects (SAEs) following COVID-19 booster vaccination: (1) a prospective observational cohort study was used to describe gender differences in SAE incidence; and (2) an unmatched nested case-control study was used to identify factors associated with SAEs among vaccinees who received the first booster dose. The study was restricted to a population vaccinated in Thailand. Overall, the study design was appropriate for the study objectives, and the study focus and findings should be of general interest to readers, given the lack of research on sex differences in SAEs associated with COVID-19 booster vaccines. I have a few comments for the authors’ consideration below.

(1)    In the Introduction, the authors make a broad statement that there is “general agreement that the primary series of COVID-19 vaccines are the most effective method for controlling the COVID-19 pandemic” (lines 33-35).  This statement should be supported by references. Similarly, a reference(s) is needed to support the statement that “females respond more effectively to vaccination” (lines 40-41). Also, are the authors talking about vaccination in general or specifically, the COVID-19 primary series?  This should be clarified.

(2)    Also in the Introduction, the effectiveness of the COVID-19 booster vaccines, as stated by the authors in line 56, should be specified.

(3)    For Figure 1 (COVID-19 timeline), the asterisk associated with the annotation on limited use should have a corresponding asterisk in the figure.

(4)    The authors should explain “Thailand’s uniqueness” (line 87) and how it is relevant to this study.

(5)    In line 92 of the Introduction, please clarify whether you are referring to a first COVID-19 booster dose versus a second.

(6)    In Figure 3 (study population), what does the asterisk after “87,615 (0.3%) received other vaccine” denote?

(7)    Table 4:  By “Crude Odds Ratio”, do the authors mean unadjusted odds ratio?

(8)    This statement is confusing and needs to be clarified:  “Also, it was indifference to older women aged 60-75” (line 286).

(9)    Statement in lines 364-365 (“…it was safe for seriousness”) is confusing as well and needs clarification.

Comments on the Quality of English Language

English needs to be improved for clarity and accuracy.

Author Response

Dear the Editor,

Thank you and the reviewers very much indeed for your kind assistance. We revise the manuscript vaccines-2632345, titled: “Sex differences in serious adverse events reported following booster doses of COVID-19 vaccination in Thailand: a countrywide nested unmatched case-control study” according to the reviewers’ comments, as follows:

We marked all changes in the main document – marked copy, please switch to mode “Simple markup” track changed or the PDF document that the system generates because we refer to line number by that mode.

  • In the Introduction, the authors make a broad statement that there is “general agreement that the primary series of COVID-19 vaccines are the most effective method for controlling the COVID-19 pandemic” (lines 33-35). This statement should be supported by references.

Follow your advice by adding reference (1): World Health Organization. WHO SAGE roadmap on uses of COVID-19 vaccines in the context of OMICRON and substantial population immunity. 2023;(January 2022). Please see line: 37

  • Similarly, a reference(s) is needed to support the statement that “females respond more effectively to vaccination” (lines 40-41). Also, are the authors talking about vaccination in general or, specifically, the COVID-19 primary series? This should be clarified.

Follow your advice by adding that it refers to “COVID-19 vaccines and some viral vaccines” when it comes to the sentence “females respond more effectively to vaccination” and adding that it refers to “both primary series and booster doses of COVID-19 vaccination” (line 41-45). We also put specific references to each topic.

  • Also in the Introduction, the effectiveness of the COVID-19 booster vaccines, as stated by the authors in line 56, should be specified.

Follow your advice, please see line 58.

  • For Figure 1 (COVID-19 timeline), the asterisk associated with the annotation on limited use should have a corresponding asterisk in the figure.

Follow your advice, please see Figure 1.

  • The authors should explain “Thailand’s uniqueness” (line 87) and how it is relevant to this study.

Follow your advice, please see line 87-90

  • In line 92 of the Introduction, please clarify whether you are referring to a first COVID-19 booster dose versus a second.

We clarify this sentence by adding “first booster dose”.

  • In Figure 3 (study population), what does the asterisk after “87,615 (0.3%) received other vaccines” denote?

We deleted it since other vaccines could be assumed after reading the text and Figure 1.

  • Table 4: By “Crude Odds Ratio”, do the authors mean unadjusted odds ratio?

Yes

  • This statement is confusing and needs to be clarified: “Also, it was indifference to older women aged 60-75” (line 286).

We deleted it because it caused more confusion than clarification.

  • Statement in lines 364-365 (“…it was safe for seriousness”) is confusing as well and needs clarification.

We clarify it in line 378-379.

We revise English language issue carefully with through rough editing from native speaker.

We submit the revised manuscripts, both a clean version and a track change version.

Please contact the corresponding authors for any future correspondence.

Your kind assistance will be deeply appreciated.  

Sincerely yours,

Chawisar Janekrongtham

Reviewer 2 Report

Comments and Suggestions for Authors

The authors analyzed the incidence of SAE following booster COVID-19 vaccines (produced by Pfizer, Moderna, and AstraZeneca) among different age and gender categories. 562 cases (male;262, and female:290) were extracted from database. More cases with SAE were reported in males aged 61-80, but conversely, more cases in females aged 21-40. In males, cardiovascular disorders were reported at 38-39% and thrombotic events were at 27-30% without age differences. Allergic reactions were frequently reported in young females and higher proportion but thrombotic events in older females. I have several comments.

1.         In table 1, they reported 152 death cases (male 98, female 54) but did not mention the detailed profiles: age, diagnosis (cardiovascular diseases, neurological diseases, exacerbation of pre-existing diseases, anaphylaxis, or unknown events).

2.         In several reports, young males were more likely to associate with myocarditis. In the present study, they reported that cardiovascular disorders were frequently in males without age differences. They discussed the higher risk of cardiovascular disorders in older males; smoking, alcohol taking, metabolic disorders such as diabetes. Authors should explain the reasons for similar incidence without age differences and it was different from the other reports, where young male was higher risk of carditis following second doses of mRNA vaccines.

3.         Authors mentioned in the Introduction, females experienced more AE and younger generations were tended to be associated with AE in several reports. In table 3, incidence of SAE was higher in males aged 61-80 years but higher in female aged >81 years. Please explain the reason (s) of difference between 61-80 years and >81 years.

4.         In figure 4, the probability of SAE was higher in older males and females than younger generation. It seemed to be quite different from other reports, where younger generations experienced more SAE. Please explain the reason(s).

5.         The present study was based on the database following the booster doses of three vaccines. Do you have any data of SAE after the primary 1st and 2nd doses in Thailand. Most reports on SAE were after the primary doses.

6.         In the conclusion line 401-404, future vaccine investments should aim to limit reactogenicity for safety concerns. No further booster recommendation is needed among younger adults without underlining diseases if a COVID-19 wave would not emerge. From their present data, SAE was reported in older generations and sentences are contradicted to limit the reactogenicity.

Author Response

Thank you and the reviewers very much indeed for your kind assistance. We revise the manuscript vaccines-2632345, titled: “Sex differences in serious adverse events reported following booster doses of COVID-19 vaccination in Thailand: a countrywide nested unmatched case-control study” according to the reviewers’ comments, as follows:

We marked all changes in the main document – marked copy, please switch to mode “Simple markup” track changed or the PDF document that the system generates because we refer to line number by that mode.

  1. In table 1, they reported 152 death cases (male 98, female 54) but did not mention the detailed profiles: age, diagnosis (cardiovascular diseases, neurological diseases, exacerbation of pre-existing diseases, anaphylaxis, or unknown events).

We could not get details of deaths by extracting these databases, including all underlying diseases and unable to confirm causal relationships of the COVID-19 vaccine. Therefore, we addressed this limitation in lines 394-395 and 402-404. The only thing we could do was sub-analysis of death outcomes of ages and SAE categories to hypothesize other results in the study.

  1. In several reports, young males were more likely to associate with myocarditis. In the present study, they reported that cardiovascular disorders were frequently in males without age differences. They discussed the higher risk of cardiovascular disorders in older males; smoking, alcohol taking, metabolic disorders such as diabetes. Authors should explain the reasons for similar incidence without age differences and it was different from the other reports, where young male was higher risk of carditis following second doses of mRNA vaccines.

Follow your advice by adding detailed clarification in the discussion part: myopericarditis cases in young males were underreported. The reasons are firstly there were too few populations at this age who got booster doses to capture these rare outcomes. Secondly, since myopericarditis has mild symptoms and fully recovered, hospital-based surveillance that was designed to capture seriousness would not capture the cases. Please see line 345-353.

  1. Authors mentioned in the Introduction, females experienced more AE and younger generations were tended to be associated with AE in several reports. In table 3, incidence of SAE was higher in males aged 61-80 years but higher in female aged >81 years. Please explain the reason (s) of difference between 61-80 years and >81 years.

According to Table 1 and 3, sparse data bias could happen in aged >81 years due to a limited sample compared to aged 61-80. Opposite to the result from logistic regression model (Figure 4), we avoid interpretation of the difference in these ages because the difference of aged >81 years is not statistically significant in Table 1. We added this limitation in line 399-403.

  1. In figure 4, the probability of SAE was higher in older males and females than younger generation. It seemed to be quite different from other reports, where younger generations experienced more SAE. Please explain the reason(s).

It was clear that most publications reported that younger generations experienced more AE but were not specific to seriousness. Younger experienced more SAE in some specific diseases, too, such as myopericarditis in young males and allergic events in young females. The main reasons could be the proportion of disease classifications and age-sex-specific organ involment following booster dose. Our study is the first study to address “overall” SAE risk following booster doses; we carefully discuss this in lines 312-324 following your comment.

  1. The present study was based on the database following the booster doses of three vaccines. Do you have any data of SAE after the primary 1st and 2nd doses in Thailand. Most reports on SAE were after the primary doses.

One previous study in Thailand (Myocarditis and Pericarditis following COVID-19 Vaccination in Thailand: https://pubmed.ncbi.nlm.nih.gov/37112661/)  uses the same database (AEFI-DDC) but is specific to Myocarditis and Pericarditis incidents after the primary series. Young males dominate the cases.

However, in our study, we could not capture this because most SAE is rarer in booster doses than primary doses, and this age group rarely got booster doses (far less study population at this specific age). Therefore, capturing rare outcomes is a challenge in booster doses. Please see line 346-349.

  1. In the conclusion line 401-404, future vaccine investments should aim to limit reactogenicity for safety concerns. No further booster recommendation is needed among younger adults without underlining diseases if a COVID-19 wave would not emerge. From their present data, SAE was reported in older generations and sentences are contradicted to limit the reactogenicity.

In the discussion part, reactogenicity is the main contribution to SAE in both older groups (lines 325-327) and younger females (line 336-340). The explanation of higher risk in AstraZeneca too (line 368-371).

We revise English language issue carefully with through rough editing from native speaker.

We submit the revised manuscripts, both a clean version and a track change version.

Sincerely yours,

Chawisar Janekrongtham

Reviewer 3 Report

Comments and Suggestions for Authors

This document is a valuable contribution to the current debate on the safety of vaccines used in the prevention of Covid-19 and specifically on the booster dose with vaccines that use mRNA and live attenuated technology. Thailand's experience merits dissemination due to the quality of its health system in the development of clinical studies and the compilation of data resulting from health interventions. The present study has been well designed and explores the main variables that have been the subject of debate in other works. The results are presented clearly, using appropriate data management. The conclusions fit the results and result from a robust discussion of their results and the state of the art. The present study contributes to the discussion on the risk-benefit of receiving a booster dose based on the analysis of the results by sex and age.

I only highlight some minor deficiencies of a formal nature. A better organization of table 3 and other spelling and English revision aspects already mentioned can be suggested, but they do not detract from this document.

Minor editing of English language required: avoid long sentences, sometimes difficult to understand. Also ortographical errors for example under 2.2 ( occupations at risk, , vaccination dates, , and vaccine brands.).  A careful revision is required.

Comments on the Quality of English Language

Minor editing of English language required: avoid long sentences, sometimes difficult to understand. Also ortographical errors for example under 2.2 ( occupations at risk, , vaccination dates, , and vaccine brands.).  A careful revision is required.

Author Response

Thank you and the reviewers very much indeed for your kind assistance. We revise the manuscript vaccines-2632345, titled: “Sex differences in serious adverse events reported following booster doses of COVID-19 vaccination in Thailand: a countrywide nested unmatched case-control study” according to the reviewers’ comments, as follows:

We marked all changes in the main document – marked copy, please switch to mode “Simple markup” track changed or the PDF document that the system generates because we refer to line number by that mode.

  1. A better organization of table 3

Follow your advice, please see Table 3.

  1. Minor editing of English language required: avoid long sentences, sometimes difficult to understand. Also ortographical errors for example under 2.2 ( occupations at risk, , vaccination dates, , and vaccine brands.). A careful revision is required.

We revise English language issue carefully with through rough editing from native speaker.

We submit the revised manuscripts, both a clean version and a track change version.

Sincerely yours,

Chawisar Janekrongtham

Reviewer 4 Report

Comments and Suggestions for Authors

Thank you for the invitation to review this manuscript. The authors have followed appropriate methods to conduct this study. However, there are few things that needed to be cleared. The conclusion is not clear about the differences in sex for reporting of SAE. The results do not state clearly about the differences of reporting or incidence of SAE among two genders. For example, the study concluded that high-risk groups of reported SAE after booster doses of COVID-19 vaccination were older males, older females, and younger females. In this phrase, can we say that females of all age groups, either young or old, are in high risk group? can we also say that older males are in high risk groups but younger males are not in high risk groups? the age 12-40 years had higher risks?????

The authors should use the word "years" always where they are mentioning the age.

Comments on the Quality of English Language

Minor edits are needed

Author Response

Dear the Editor,

Thank you and the reviewers very much indeed for your kind assistance. We revise the manuscript vaccines-2632345, titled: “Sex differences in serious adverse events reported following booster doses of COVID-19 vaccination in Thailand: a countrywide nested unmatched case-control study” according to the reviewers’ comments, as follows:

We marked all changes in the main document – marked copy, please switch to mode “Simple markup” track changed or the PDF document that the system generates because we refer to line number by that mode.

  1. The conclusion is not clear about the differences in sex for reporting of SAE. The results do not state clearly about the differences of reporting or incidence of SAE among two genders. For example, the study concluded that high-risk groups of reported SAE after booster doses of COVID-19 vaccination were older males, older females, and younger females. In this phrase, can we say that females of all age groups, either young or old, are in high risk group? can we also say that older males are in high risk groups but younger males are not in high risk groups? the age 12-40 years had higher risks?????

We decided to explain the sex difference in discussion part that “This study also found that age modified the effect of sex. Females aged 12-40 had a higher risk of reported SAE than males of the same age, while males over 50 had higher risk than females. Please see line 309-311. We would like to avoid concluding that younger males are not at high risk due to a limitation of the study sample; we carefully explain it in line 345-353. However, older males, older females, and younger females had a higher risk of SAE than younger males in our study with the supporting mechanisms that link to host-specific responses (line 325-344). Therefore, we conclude by the mechanism that  “sexes responded differently to the vaccines. Mechanisms could be explained by individual host response rather than vaccines' direct effect, and as a result, SAE could be preventable”. Please see line 416-418.

  1. The authors should use the word "years" always where they are mentioning the age.

Follow your advice in some words after consultation with native-speaker.

We revise English language issue carefully with through rough editing from native speaker.

We submit the revised manuscripts, both a clean version and a track change version.

Sincerely yours,

Chawisar Janekrongtham

Round 2

Reviewer 2 Report

Comments and Suggestions for Authors

In the revised manuscript, authors responded adequately to my comments. The data shows the gender differences in the occurrence of  severe adverse events following COVID-19 vaccines.

I have no additional comment.